



**Exploring the potential of utilizing high resolution X-band radar for**
**urban rainfall estimation**
Wen-Yu Yang[1], Guang-Heng Ni[1], You-Cun Qi[2,3], Yang Hong[1,4], Ting Sun[1*]
*1) State Key Laboratory of Hydro-Science and Engineering, Department of Hydraulic*
*Engineering, Tsinghua University, Beijing 100084, China*
*2) Cooperative Institute for Mesoscale Meteorological Studies, University of Oklahoma*
*3) NOAA/OAR/National Severe Storms Laboratory, Norman, Oklahoma*
*4) Department of Civil Engineering and Environmental Science, University of*
*Oklahoma, Norman, Oklahoma*
* Corresponding Author: sunting@tsinghua.edu.cn



**Abstract:**
X-band-radar-based quantitative precipitation estimation (QPE) system is increasingly
gaining interest thanks to its strength in providing high spatial resolution rainfall
information for urban hydrological applications. However, prior to such applications, a
variety of errors associated with X-band radars are mandatory to be corrected. In
general, X-band radar QPE systems are affected by two types of errors: 1) common
errors (e.g. mis-calibration, beam blockage, attenuation, non-precipitation clutter,
variations in the raindrop size distribution) and 2) "wind drift" errors resulting from
non-vertical falling of raindrops. In this study, we first assess the impacts of different
corrections of common error using a dataset consisting of one-year reflectivity
measurements collected at an X-band radar site and a distrometer along with rainfall
measurements in Beijing urban area. The common error corrections demonstrate
promising improvements in the rainfall estimates, even though an underestimate of 24.6%
by the radar QPE system in the total accumulated rainfall still exists as compared with
gauge measurements. The most significant improvement is realized by beam
integration correction. The DSD-related corrections (i.e., convective–stratiform
classification and local $Z$-$R$ relationship) also lead to remarkable improvement and
highlight the necessity of deriving the localized $Z$-$R$ relationships for specific rainfall
systems. The effectiveness of wind drift correction is then evaluated for a fast-moving
case, whose results indicate both the total accumulation and the temporal characteristics
of the rainfall estimates can be improved. In conclusion, considerable potential of X-
band radar in high-resolution rainfall estimation can be realized by necessary error



corrections.
**Keywords:** urban hydrology, X-band radar, quantitative precipitation estimation, error
correction, wind drift effect


## 1. Introduction

Urban flash flooding is one of the most severe hazards in cities (Schmitt et al., 2004; Yang et al., 2015a). Large coverage of impervious surfaces in cities will exaggerate the flooding since heavy rainfall is more likely to transform into runoff instead of infiltrating into the soil. To mitigate its detrimental effects, accurate prediction of runoff at high spatiotemporal resolution is critical for emergency management and warning operations. When conducting the hydrological and/or hydraulic simulations, the spatiotemporal variability of rainfall is known to be the major source of a range of uncertainties (Schellart et al., 2012; Schröter et al., 2015; Rafieeinasab et al., 2015; Rico-Ramirez et al., 2015), which thus warrants compelling need for high-resolutions rainfall data (Schilling, 1991; Emmanuel,et al, 2012; Eldardiry et al., 2015).

Weather radars have been worldwide recognized as essential tools to provide high-resolution rainfall measurements (Smith and Krajewski, 1991; Krajewski and Smith, 2002; Li et al., 2014; Li et al., 2015). The current operational weather radar systems (e.g., NEXRAD in America, OPERA in Europe) based on C-band and/or S-band radars operating on long-range coverage. However, their spatiotemporal resolutions of 1 km/5–10 min are insufficient to support accurate estimations of precipitation variability in urban area (Krajewski et al., 2002; Smith et al., 2007). For instance, recent studies suggest that for urban areas less than 1 ha, rainfall input is required at the spatial resolution of ~100 m; while for urban areas between 1 ha and ~100 ha, the required resolution is relaxed to 500 m (Faures et al., 1995; Berne et al., 2004). Considering the



high utility in monitoring rainfall in urban area, X-band radars are being deployed in a
number of hydrometeorological applications (Chen and Chandrasekar, 2015).

Prior to the application of radar-based rainfall product in hydrological simulations,
quantitative precipitation estimation (QPE) systems are mandatory to be established,
where multiple error sources should be appreciated (e.g. Krajewski and Smith, 2002;
Villarini and Krajewski, 2010). One of the error sources is associated with the
reflectivity measurement, which can be attributable to mis-calibration, beam blockage,
attenuation, non-precipitation clutter, and vertical profile of reflectivity (VPR). These
errors may reduce the accuracy of reflectivity measured by weather radar (Germann et
al., 2006; Hazenberg et al., 2011a). The variability of $Z$–$R$ relationship is another
important error source. The standard $Z$–$R$ relationship (Chapon et al., 2008; Hazenberg
et al., 2011b) takes the form of $Z = aR^b$ (Marshall and Palmer, 1948), in which the
parameters $a$ and $b$ depend on the raindrop size distribution (DSD). Inappropriate
determination of $a$ and $b$ will introduce errors into the estimated rainfall. The above
errors are termed as **common errors** in this study as they have to be corrected for most
operational radars.

Besides the common errors, the rainfall estimates for X-band radar can be affected by
other weather-related dynamic processes. For instance, the falling paths of raindrops
are not perfectly vertical due to wind, implying a horizontal displacement may exist
between the aloft measurement position and ground falling location of a raindrop, or



the **"wind drift"**. The wind drift can lead to inconsistency between the estimated and
actual rainfall fields at the ground level (Fabry et al., 1994; Liu and Krajewski, 1996;
Sandford., 2015; Seo and Krajewski, 2015). Usually the wind drift is ignored for the S-
band and C-band radars due to their relatively coarse spatial resolution. However, this
effect can be remarkable for X-band radars given its high spatial resolution of ~100 m,
in particular under windy conditions that are common for convective rainfall events
(Sandford, 2015; Seo and Krajewski, 2015). As such, errors due to the wind drift should
be appreciated in the application of X-band radars in urban hydrometeorology.

In order to improve the quality of X-band radar based QPE systems in urban
hydrometeorological applications, different procedures have been explored to reduce
the aforementioned errors. Current research on X-band radar shows that using the
differential phase shift can reliably resolve the attenuation which is the primary
disadvantage of X-band radar (Anagnostou et al., 2004, 2006a,b; Park et al., 2005;
Kalogiros et al. 2014). Specifically, Kalogiros et al. (2014) developed an algorithm to
correct the attenuation of horizontal-polarization reflectivity by using an iterative
optimal parameterizations of specific differential attenuation and backscattering phase
shift. Besides the attenuation, Van de Beek et al. (2010) suggested that the effects of
non-precipitation clutter should also be carefully addressed for applications of X-band
radars in urban areas. Lo Conti et al. (2015) investigated the effect of calibration in the
$Z$–$R$ relationship of an X-band radar in Palermo (Italy), suggesting the high variability
of the $Z$–$R$ relationships determined for specific events limited its wide applicability.





Matrosov et al. (2016) found no bright band rainfall has distinct *Z-R* relationship from
those of other rain types and should be distinguished from stratiform rainfall. For
operational applications, Anagnostou et al. (2010) showed that adjusting the *Z–R*
relationship for mean-field bias with the $K_{DP}$-R estimates as reference is a promising
technique for acquiring unbiased high resolution radar-rainfall estimates. Maki et al.
(2010) developed one X-band radar QPE system for three major metropolitan areas in
Japan, which may still underestimate the rainfall by ~20% compared with the gauge
measurement even though the effects due to non-precipitation clutter, beam blockage
and attenuation have been accounted for. Chen and Chandrasekar (2015) developed a
high-resolution (250 m/1 min) QPE system in Dallas–Fort Worth urban area consisting
of polarimetric Weather Surveillance Radar 88 Doppler (WSR-88D) and X-band radars,
which demonstrated low overall biases and normalized standard errors in rainfall
accumulation products of different temporal scales (5-60 min). Although these studies
demonstrate the effectiveness of different measures in improving the X-band-radar-
based QPE systems, the relative importance of different error sources contributing to
the overall errors remains to be unknown. Furthermore, given the errors of a specific
radar QPE system strongly depend on the spatiotemporal characteristics of the local
rainfall system, an analysis of long-term rainfall characteristics is expected to enable a
better understanding of the potential of X-band-radar-based QPEs in urban
hydrometeorological applications.

In this study, we use a dataset consisting of one-year measurements by an X-band radar





in Beijing to explore its potential in high resolution rainfall estimation. The study period
extends from July 2014 to September 2015 with 43 rainfall events. Specifically, we aim
to answer the following two questions: 1) What is the relative importance of different
error sources for the overall accuracy of a QPE in urban area; 2) What are the impacts
of wind drift on the ground-level rainfall estimation?

The paper is organized as follows. Section 2 describes the study area and the dataset.
Section 3 details the correction procedures for common error sources and wind drift in
X-band-radar-based QPEs. The relative importance of correcting each common error
for rainfall estimation is discussed in Section 4, followed by a case study to show the
implication of wind drift correction. Finally, the concluding remarks are provided in
Section 5.

**2.  Study Area and Data Description**
Beijing, the capital city of China with more than 21 million residents, features complex
topography, with mountains to its north and west and a highly urbanized area in its
eastern part. Beijing is prone to summertime heavy rainfall events that occasionally
lead to severe flash floods (Yang et al., 2014b).

The dataset used in this study consists of one-year measurements by a single-polarized
X-band radar (Fig. 1a, hereafter referred as Beijing Radar) in the northwest of Beijing.
Technical specifications of the Beijing X-band radar are given in Table 1. A full



volumetric scanning by the Beijing Radar is performed every 7 min at 14 elevations
(0.5 °, 0.9 °, 1.3 °, 1.8 °, 2.4 °, 3.1 °, 4.0 °, 5.1 °, 6.4 °, 8, 10.0 °, 12.0 °, 15.6 °, and 19.5 °). Each
scan has 400 gates along the beam with a gate resolution of 90 m and a maximum range
of 36 km.

An OTT Parsivel disdrometer (Fig.. 1b) deployed near the Beijing Radar (within 5 km,
cf. Fig. 2) and a gauge network consisting of 8 standard tipping-bucket gauges (Fig. 2)
are used to validate the QPE system for Beijing Radar. The disdrometer can archive 32
equivalent diameter classes (ranging from 0 to 26 mm with varying diameter increments
between 0.125 and 3 mm) and 32 different velocity classes at 1-min resolution. Other
specifications of the disdrometer are provided in Table 2.

In order to avoid the temporal bias among radar, disdrometer and rain gauge
measurements, all the measurements are conformed at 1-h resolution for subsequent
analysis. We note that the bias correction by assimilating gauge data were not
performed in this study due to its shadow effect over the corrections in radar QPE
system (Hazenberg et al., 2011a).

**3. Rainfall estimation algorithm**
**3.1 Procedures for common error corrections**
The common errors introduced in reflectivity measurement are mostly due to radar
miscalibration, non-precipitating echo contamination, signal attenuation, beam



174 blockage and VPR, whereas the errors in *Z-R* conversion rely on the variability in *Z-R*

175 relationships for different rainfall types. We note that the errors due to VPR are not

176 considered for two reasons: 1) the lack of required a priori knowledge of vertical profile

177 of rainfall in Beijing and 2) the minimal influence of VPR in this study since the

178 maximum effective measurement height of 2.5 km of Beijing Radar is lower than the

179 melting layer height of ~4 km for the warm-season rainfall (Cao and Qi, 2014).

180

181 **3.1.1 Radar calibration**

182 Radar calibration is conducted to fix the inappropriate parameter settings. Although the

183 Beijing Radar has been calibrated before installation, post-installation calibration

184 should be conducted to cancel the possible aging and thermal effects (Collier, 1989).

185 Among all the available calibration approaches (Manz et al., 2000), calibration with

186 disdrometer data would be the most straightforward one (Lee and Zawadzki, 2006).

187 With the help of a nearby disdrometer (cf. Fig. 2 for its location), Beijing Radar is

188 calibrated by comparing the radar measurements with disdrometer measurements.

189 The disdrometer-based estimate of *Z* can be expressed as (Delrieu et al., 1999):

$$Z = \frac{10^6 \lambda^4}{\pi^5 |K|^2} \int_0^\infty \sigma_B(D) N(D) dD, \qquad (1)$$

190 where $\lambda$ is the operational wavelength of the radar (3.21 cm in this study), $|K|^2$ a

191 coefficient dependent on the dielectric constant of water. Assuming a raindrop is a

192 sphere of diameter *D* (cm), DSD spectra $N(D)$ ($cm^{-4}$) is defined as the raindrop

193 concentration in a given air volume as a function of *D* and $\sigma_B(D)$ is the backscattering



cross-section area ($cm^2$) as a function of $D$. Note that the Mie calculation is adopted in
this study since the Rayleigh approximation does not satisfy the condition $D \leq \lambda/16$
when $\lambda = 3.21$ cm.

The comparison of the reflectivity measurement between radar and disdrometer
demonstrates overall consistency with observed underestimates by radar for reflectivity
exceeding 35 dBZ (an example for the comparison of the 1[th] September 2014 event is
shown in Fig. 3). Except for the calibration error, this underestimation may be the joint
effect of attenuation and inconsistency between the ground-level point measurements
and the aloft volume measurements. Therefore, although there is a calibration drift for
the Beijing Radar, it was decided not to apply any correction to the reflectivity
measurements for the 43 selected events.

**3.1.2 Non-precipitating echo removal**
Radar echoes may be contaminated by non-precipitating echoes that need to be
identified and removed before rainfall estimation. Ground clutters and anomalous
propagations are the two main sources of non-precipitating echoes (Steiner and Smith,
2002). Ground clutters are caused by scattering in the antenna sidelobes hitting the
ground close to the radar site as well as by fixed objects (Villarini and Krajewski, 2010).
Anomalous propagations are contamination of radar reflectivity data from echoes
normally not seen by the radar. In particular, anomalous propagations remain a serious
problem for situations when they are embedded in precipitation echoes (Steiner and





Smith, 2002).

For Beijing Radar, the ground clutter effect is corrected by removing the echoes with
radial Doppler velocity close to zero. As the more advanced polarimetric techniques
cannot be applied to the measurements by the Beijing Radar, the well-recognized
Steiner and Smith (2002) algorithm is used for detecting anomalous propagations (e.g.,
Hazenberg et al., 2011a, Hazenberg et al., 2014). This algorithm utilizes the three-
dimensional reflectivity structure and builds upon three key parameters: the vertical
extent of radar echoes, their spatial variability and vertical gradient of intensity, which
are available in the Beijing Radar measurements.

**3.1.3 Attenuation correction**
Radar signals can be attenuated during propagation (Atlas and Banks, 1951), in
particular for X-band radars running at short wavelengths. For single-polarized radars,
correction algorithms can be categorized into forward and backward algorithms
(Delrieu et al., 1999). Among the forward algorithms, the HB algorithm (Hitschfeld
and Bordan, 1954) is widely applied and thus adopted in this study for attenuation
correction, whose key steps are recapitulated here.

For a well calibrated radar without blind-range attenuation (e.g. radome attenuation),
its measured reflectivity $Z_m$ can be expressed as:

$$Z_m(r) = Z(r) A(r), \qquad (2)$$



where $Z(r)$ is the true reflectivity at the same range and $A(r) =$
$\exp\left(-\frac{2\,ln(10)}{10}\int_0^r k(s)ds\right)$ is the path-integrated attenuation (PIA) factor with the
specific attenuation $k(s)$ at a distance $s$ being the only unknown to resolve.

Meanwhile the relationship between the true reflectivity $Z$ and the specific attenuation
$k$ can be given by:

$$Z = ck^d, \tag{3}$$

where $c$ and $d$ are the DSD related parameters. Therefore, by canceling $k$, the
relationship between $Z_m(r)$ and $Z(r)$ can be obtained as follows:

$$Z(r) = \frac{Z_m(r)}{\left(1 - \frac{2\,ln(10)}{10}\int_0^r \left(\frac{Z_m(s)}{c}\right)^{\frac{1}{d}} ds\right)^d}, \tag{4}$$

where the parameters $c$ and $d$ can be determined by $Z$-$k$ regression (Eq.3) with
distrometer measurements. To avoid the numeric instability known in the HB algorithm,
a maximum PIA of 10 dB is specified in this study.

Also, knowing the DSD, $k$ can be calculated by:

$$k = c_k \int_0^\infty Q_t(D)N(D)dD, \tag{5}$$

where $D$ (cm) denotes the raindrop diameter, $Q_t$ (cm$^2$) the Mie total attenuation cross
sections (refer to Delrieu et al. (1999) for the calculation method of $Q_t$), $N$ (cm$^{-4}$) the
raindrop concentration in a given air volume and $c_k = 0.4343 \times 10^6$ a constant.

By conducting $Z$-$k$ regression (Eq. 3) with distrometer measurements from July 2014



to September 2015 without differentiating rainfall types (Fig. 4), the required
parameters for $Z_m$-$Z$ relationship (Eq.4) are determined as $c = 1.12 \times 10^5$ and $d =$

257   1.1.


**3.1.4 Beam Integration**
Due to the presence of obstructions on the beam path in terrain-complex contexts (e.g.
urban, mountainous areas), beam blockage frequently occurs and thus compromise the
accuracy of radar QPE. In particular, considering the radar are usually operated at the
low elevation angles where more ground clutters exist, removal of beam blockage effect
is of great importance for radar QPE systems. As such, beam integration, a technique
to avoid the beam blockage by integrating measurements at different elevations for
different azimuths, is conducted in this study.

In this study, the starting integral elevation, or the lowest optimal elevation, for a
specific azimuth range is determined as the lowest elevation that at which the beam
blockage of all ranges in this azimuth range must be less than 50% (cf. Fig. 2 for the
starting integral elevations in this study). For instance, measurements at the elevation
of 4.0 ° for the azimuth between 177 ° and 181 ° are used in this study since two tall
buildings stand to the south of the Beijing Radar.

**3.1.5 Convective–stratiform classification**
Because of the distinct DSD characteristics between the convective and stratiform





rainfall, the *Z-R* relationships for the two types of rainfall differ significantly. It is thus
necessary to conduct the convective-stratiform classification before the *Z-R* conversion.
Due to the unavailability of real-time atmospheric temperature profiles that is
commonly used for convective-stratiform classification, a vertically integrated liquid
(VIL; Greene and Clark, 1972) based method (Zhang and Qi, 2010) is adopted in this
study.

With the volume scan reflectivity measurements, VIL can be obtained by:

$$\mathrm{VIL} = \sum_k \mathrm{VILpar}_k, \qquad (6)$$

where $\mathrm{VILpar}_k = \mathrm{LW} \cdot \mathrm{DB}$ denotes the VIL within the $k^{\mathrm{th}}$ tilt with LW (kg km$^{-3}$) and
DB (km) being the liquid water content associated with a particular value of reflectivity
and the depth of a radar beam, respectively, given by

$$\mathrm{LW} = 3.44 \times 10^3 Z^{\frac{4}{7}}, \qquad (7)$$

$$\mathrm{DB} = \begin{cases} \mathrm{BH}\left[\theta_{k_{top}} + 0.5\mathrm{BW}\right] - \mathrm{BH}\left[0.5\left(\theta_{k_{top}} + \theta_{k_{top}-1}\right)\right] & k = k_{top} \\ \mathrm{BH}[0.5(\theta_{k+1} + \theta_k)] - \mathrm{BH}[0.5((\theta_k + \theta_{k-1}))] & 1 < k < k_{top}, \quad (8) \\ \mathrm{BH}[0.5((\theta_1 + \theta_2))] & k = 1 \end{cases}$$


where DB (km) is the depth of a radar beam, *Z* the radar reflectivity within a radar
sample volume, BW the angular width of the radar beam between the half-power points,
BH the beam center height for a given elevation angle and range under the standard
atmospheric refraction conditions and $\theta_k$ the elevation angle at the $k^{\mathrm{th}}$ tilt.

Once the knowledge of VIL is obtained, the reflectivity pixels can be categorized into
stratiform with VIL $< 6.5$ kg m$^{-2}$ and convective with VIL $\geq 6.5$ kg m$^{-2}$.






### 3.1.6 Derivation of local *Z-R* relationships

The standard *Z-R* relationships are $Z = 200\ R^{1.6}$ and $Z = 300\ R^{1.4}$ for stratiform and convective situations, respectively. However, due to the large variability of *Z-R* relationship among different locations, localized *Z-R* relationships are mandatory for building radar QPE systems. As such, we use the DSD measurements collected at the distrometer sites near the Beijing Radar to establish the local *Z-R* relationships.

As can be seen in Fig. 5, by conducting *Z-R* regression without differentiating rainfall types, $Z=428.4\ R^{1.2}$ is obtained from non-linear regression. Furthermore, considering the rainfall types (stratiform for $Z \leq 39$ dBZ and convective for $Z > 39$ dBZ; cf. Steiner et al., 1995), *Z-R* relationships $Z=426.5\ R^{1.3}$ and $Z=499.3\ R^{1.2}$ are obtained for stratiform and convective event, respectively. It is noteworthy that the *Z-R* relationships derived by this study distinguish from the standard forms, implying the necessity of locally-derived forms in radar QPE systems. It is also noteworthy the convective events comprise a large portion of the rainfall events, which is consistent with the previous findings that convective events are common in urban areas. (Yang et al., 2014a, Yang et al., 2015b, Yu et al., 2015).

### 3.2 Procedure for wind drift correction

Wind drift means the horizontal displacement between the aloft measurement position and ground falling location of a raindrop. Wind drift correction enables better rainfall





estimation at the ground level. The horizontal displacement $\Delta x$ of a raindrop can be
estimated by integrating horizontal wind velocity $u(h)$ over falling duration (Caroline,
2015) as follows:

$$\Delta x = \int_0^t u(h)dt = \int_0^{h_b} \frac{u(h)}{w(h)} dh, \qquad (9)$$

where $w(h)$ is the falling speed at height $h$ and $h_b$ is the height of the radar beam at the
measurement location. And $u(h)$ is given by:

$$u(h) = S \cdot h, \qquad (10)$$

where $S$ is a constant wind shear (Caroline, 2015).
By assuming a zero wind speed at the ground level, the wind shear $S$ can be calculated
by:

$$S = \frac{u(h_b)}{h_b}. \qquad (11)$$

Given the constant falling speed of 5 m s$^{-1}$ for raindrops blow the melting layer
(Caroline 2015), the Eq. (12) thus can be simplified as:

$$\Delta \mathrm{x} = \frac{1}{5} \frac{S h_b^2}{2} = \frac{u(h_b) h_b}{10}. \qquad (12)$$

Furthermore, $u(h_b)$ can be determined by a pixel-based tracking algorithm for short-
term quantitative rainfall forecasting (Zahraei et al., 2012). The tracking algorithm can
estimate the advection velocity of a rainy pixel (equal to the background wind velocity)
by tracking its successive positions between radar images based on the maximum
correlation of meshes in consecutive images.

**4. Results and discussion**
**4.1 Importance of different common corrections**





To assess the relative importance of different common error corrections, a rotation-
based strategy is conducted, which consists of two procedures as follows:
1) **Complete-correction (CC) procedure**: all the corrections described in Section 3.1

are applied to the reflectivity measurements to obtain the rainfall field as the best

estimate;

2) **Partial-correction (PC) procedure**: all but one of the corrections as in the CC

procedure are used to estimate the rainfall, whereby the corresponding estimate can

be compared with the best estimate of CC procedure to assess the effectiveness of

a specific error correction (i.e., the excluded correction). By rotating the specific

correction in the PC procedure, rainfall estimates without different corrections can

thus be obtained.


Furthermore, to quantify the correction effectiveness so that get the relative importance
of different error sources contributing to the overall errors, the radar-gauge ratio of daily
accumulated rainfall $rd$, average radar-gauge ratio of total rainfall $ra$, the root-mean-
square error RMSE, and the coefficient of determination $r^2$ are introduced as follows:

$$rd_{i,k} = \sum_n R_{n_{i,k}}/\sum_n G_{n_{i,k}}, \tag{13}$$

$$ra = \frac{1}{8}\sum_{k=1}^{8}\sum_n R_{n,k}/\sum_n G_{n,k}, \tag{14}$$

$$\text{RMSE} = \sqrt{\frac{1}{N}\sum_{n=1}^{N}(R_n - G_n)^2}, \tag{15}$$

$$r^2 = \frac{\left(\sum(R_n - \overline{R_n})(G_n - \overline{G_n})\right)^2}{(\sum_{n=1}^{N}(R_n - \overline{R_n})^2)(\sum_{n=1}^{N}(G_n - \overline{G_n})^2)} \tag{16}$$

where the subscript $i$ and $k$ denote the date number and gauge number, respectively;
while the radar-based estimates $R$ and gauge measurements $G$ with n and N denoting



the n-th hour and total number of hours, respectively. The first two metrics are chosen
to describe the systematic bias, while the last two assess the average error magnitude
and agreement at hourly scale.

To reduce the bias in the radar-gauge comparison, 33 events, during which at least three
gauges have valid measurements, are chosen and investigated for the correction
effectiveness. For the 33 events, the ratio $rd$ based on the CC procedure varies mostly
between 0 and 3 with the medians ranging between 0.5 and 1.5 (Fig. 6), suggesting a
promising performance of common corrections, though considerable variability can be
observed for several events (e.g., events of July 29, 2015). Extending beyond the 33
chosen events to all the events in the study period, it is also noted that with the CC
procedure performed, the Beijing Radar underestimates the rainfall compared with the
gauges (cf. a linear radar-gauge regression slope of 0.69 and coefficient of
determination $R^2$ of 0.76 in Fig. 7) with an averaged underestimate of 24.6% in the total
rainfall. This result is comparable with that of the X-band radar at the Delft University
(cf. a linear radar-gauge regression slope of 0.65 reported in Van de Beek et al. (2010)).

The influence of the PC procedure on the rainfall estimates is then examined by
comparing the above metrics between the radar-based estimates and gauge
measurements during the study period (Table 3 and Fig. 8). Correction for anomalous
propagations contributes a minimal improvement in the rainfall estimates (Fig. 8a)
because this correction may reduce the estimated reflectivity and thus the rainfall.



However, such minimal influence does NOT suggest the non-necessity of anomalous
propagation correction since its improvement is largely screened by errors from other
sources. The attenuation correction demonstrates improvement in the rainfall estimates
in terms of both $ra$ and RMSE (Fig. 8b). As expected, the largest improvement is
resulted from the beam integration as indicated by the reduced RMSE (cf. RMSE from
3.18 for -BI to 1.95 for CC in Table 3) and by the increased coefficient of determination
$r^2$ for the radar-gauge linear regression (cf. from 0.54 to 0.76 in Fig. 8c). The increased
deviation for no beam integration suggests that larger $ra$ (cf. 0.87 for -BI in Table 3) is
actually the results of offset of positive and negative deviations.

Furthermore, significant improvements are observed in the estimated rainfall when the
variability in the DSD is taken into account: both the convective–stratiform
classification and the local $Z\text{-}R$ relationship derivation contribute to increases in $ra$ and
decreases in RMSE (Table 3). Such improvements are also demonstrated by the
increases in the slope of radar-gauge linear regression (Fig. 8d and e).

In general, the DSD-related corrections (i.e., convective–stratiform classification and
the local $Z\text{-}R$ relationship) demonstrates greater improvement in the estimation of
rainfall as compared with some of the reflectivity-related corrections (i.e., anomalous
propagations removal, attenuation correction), implying the importance of appropriate
DSD-related corrections.



**4.2 Case study: a fast-moving event of 20150904**

Although considerable improvements can be observed in the rainfall estimates after the

common error corrections, it should be noted the potential for improving the rainfall

estimates may be realized by correcting the wind drift errors in radar QPE systems, in

particular for X-band radars of high spatial resolutions. To examine the effect on the

radar based rainfall estimates of wind drift correction, a long-duration event (~16 h) on

4$^{th}$ September 2015 featuring fast-moving storms with complex structures is analyzed

here.

The event originated from low pressure system passing over Beijing from southwest

towards northeast, where a high pressure zone resided over the urban area of Beijing.

Taking an episode of ~30 min for instance, several convective cells rapidly grew in the

southwest of Beijing (Fig. 9a and b), then quickly moved to northeast (Fig. 9c), and

developed to a widespread precipitating system afterwards (Fig. 9d).

The influence of the wind drift correction on the hourly rainfall estimates is then

examined by a radar-gauge linear regression (Fig. 10). After the wind drift correction,

the slope and coefficient of determination $R^2$ of the radar-gauge regression increase

from 0.46 to 0.58 and from 0.69 to 0.79, respectively, suggesting an evident

improvement in the estimates of total rainfall. In addition, the temporal characteristics

of rainfall estimates are refined as well. For instance of a gauge close to the radar, the

correlation coefficient of the rainfall series between the radar and gauge is increased



from 0.79 to 0.86 and the RMSE of rainfall measurements is reduced from 27.4 to 18.8
after the wind drift correction (Fig. 11). It is also noteworthy that the wind drift
correction leads to an improved temporal consistency in the rainfall time series between
the gauge and radar with the more accurate rainfall estimate at the peaking time (i.e.,
17:00 shown in Fig. 11).

**5. Concluding remarks**
In this study, we analyzed 43 rainfall events between July 2014 and September 2015
based on the measurements gathered by an X-band single-polarized radar, a
distrormeter and 8 rain gauges in Beijing urban area. These measurements allow us to
explore the potential for high-resolution rainfall measurement with X-band radar over
complex urban region. The impacts and importance of common corrections (i.e., radar
calibration, non-precipitating echo removal, attenuation correction, beam integration,
convective–stratiform classification and local *Z-R* relationships) on the quality of the
radar QPE is first studied, followed by an assessment of the effectiveness of wind drift
correction. The major findings are summarized as follows:
1) Although the radar QPE system underestimates the total rainfall accumulations by
24.6% as compared with gauges, the common corrections demonstrate promising
improvements in the rainfall estimates.
2) The greatest improvements in the radar based rainfall estimates can be attributed to
the beam integration, which significantly outperforms the operation at the single
lowest elevation without beam blockage (i.e., 4.0 °in this study). It is thus highly



suggested to conduct volumetric reflectivity measurements with the X-band radar
in particular for complex-terrain regions. Minor improvements on the radar rainfall
estimation are observed after the anomalous propagations removal as compared to
other common error corrections.
3) The DSD-related corrections (i.e., convective–stratiform classification and local Z-
R relationship) lead to significant improvements in the rainfall estimates. And the
local Z-R relationship plays a more crucial role in improving the rainfall estimates
compared with the other corrections, which highlights the necessity of deriving the
localized Z-R relationships for different types of rainfall.
4) The wind drift correction improve both the total accumulation and the temporal
characteristics of the estimated rainfall, which suggest the necessity of wind drift
correction for X-band radars of high spatial resolutions.

The possible future improvement relies on the inclusion of vertical profile of reflectivity
(VPR) measurements of this region to correct the underestimation in reflectivity (Qi et
al., 2013). In addition, polarimetric radars, featuring the ability to capture two-
dimensional structure of rainfall, can provide new insight into rainfall microphysics and
make further improvements in monitoring urban rainfall.


**Acknowledgement**
This work is supported by the National Science Foundation of China under grant no.



51190092 and 51409147, by the Ministry of Science and Technology of China under
Grant No. 2013DFG72270, and by China Postdoctoral Science Foundation under grant
no. 2015T80093. We are grateful to the Beijing Water Authority for the assistance in
providing the rain gauge data.





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



**Table 1 Main specifications of X-band radar used in this study.**

| Parameters | Value |
|---|---|
| Frequency | 9.38 Ghz |
| Peak transmitted power | 25 kw, 38 dB gian |
| Antenna | 1.3 m |
| Beam width | 1.8 ° |
| Platform | 44 m above ground level |

**Table 2 Specifications of disdrometer used in this study.**

| Parameters | Value |
|---|---|
| Optical sensor wavelength | 780 nm |
| Particle size range | 0.2–5 mm (liquid precipitation), 0.2–25 mm (solid precipitation) |
| Particle velocity range | 0.2–20 m s$^{-1}$ |
| Measurement time aggregation interval setting | 1 min |
| Precipitation intensity range | 0.001–1200 mm h$^{-1}$ |
| Radar reflectivity range | 9.9–99 $\pm$20% dBZ |

**Table 3 Impacts of the correcting different common errors on hourly accumulated rainfall.**

| Statistic[1] | CC procedure | PC procedure[2] | | | | |
|---|---|---|---|---|---|---|
| | | - AP | -Att | -BI | -Seg | -Z-R |
| *ra* | 0.75 | 0.77 | 0.72 | 0.87 | 0.57 | 0.61 |
| RMSE | 1.95 | 1.97 | 2.02 | 3.18 | 2.17 | 2.34 |

Notes:

1. The statistics are calculated with hourly rainfall estimates. $ra$ is the average radar-gauge

   ratio for 8 gauges and $\mathrm{RMSE} = \sqrt{\frac{1}{N}\sum_{n=1}^{N}(R_n - G_n)^2}$ is the root-mean-square error

   between the radar-based estimates $R$ and gauge measurements $G$ with $n$ and $N$ denoting

   the $n$-th hour and total number of hours, respectively.

2. The minus sign (-) in PC procedure indicates the exclusion of a specific correction. The

   correction names are simplified as follows: AP for anomalous propagations correction, Att

   for attenuation correction, BI for beam integration, Seg for convective–stratiform rainfall



classification, *Z-R* for local *Z-R* relationship derivation.

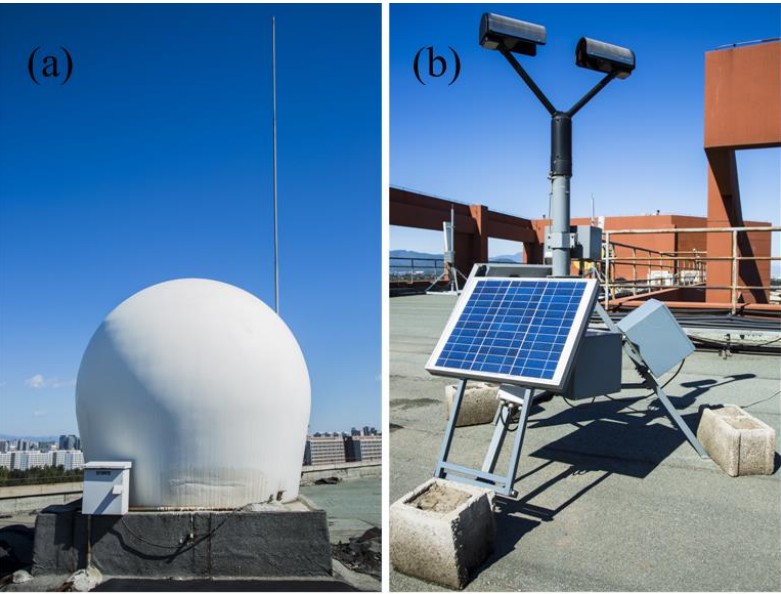


**Figure 1** The site view of (a) the Beijing Radar and (b) the disdrometer used in this
study.





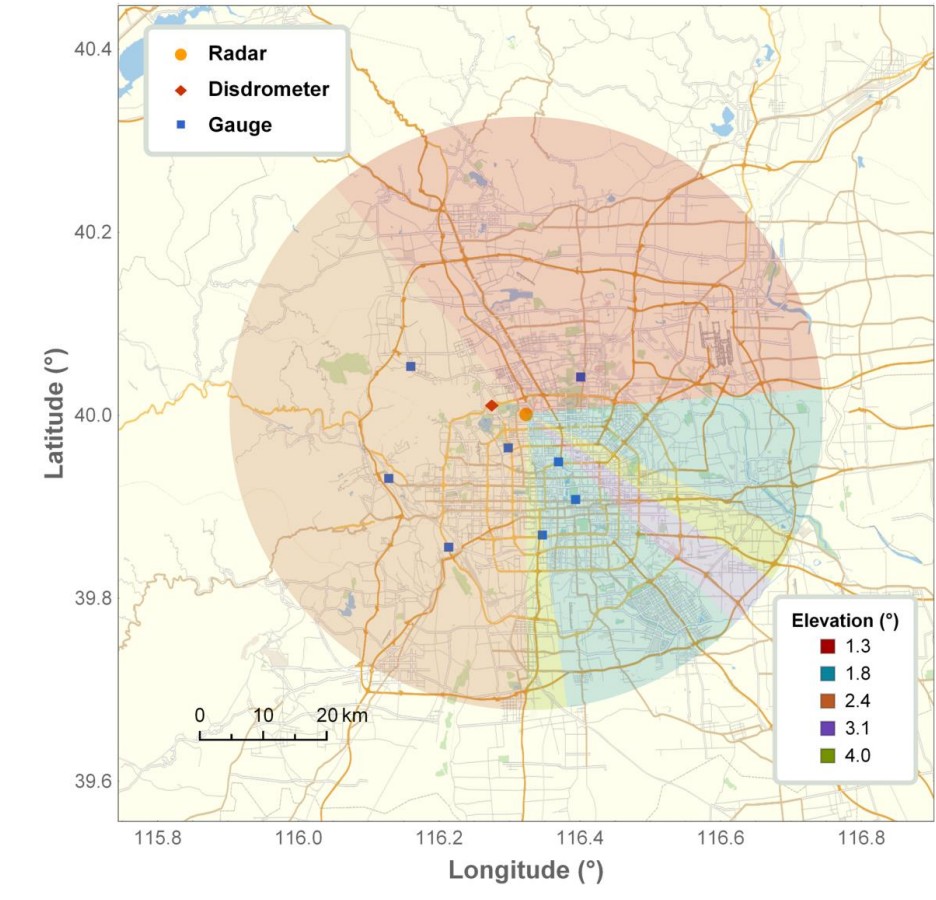


**Figure 2** The instrumentation layout of urban rainfall monitoring system of in Beijing.





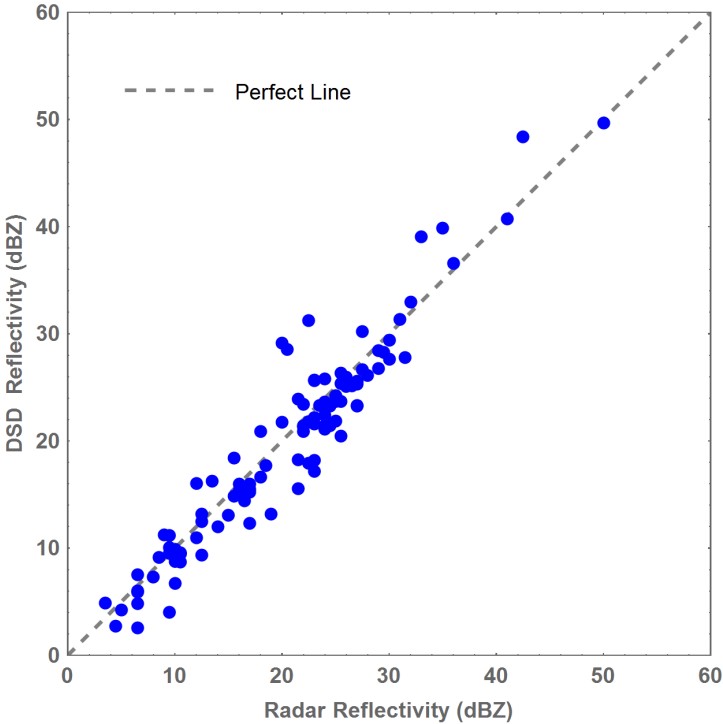


**Figure 3** The relationship between radar-measured reflectivity and distrometer-

estimated reflectivity.





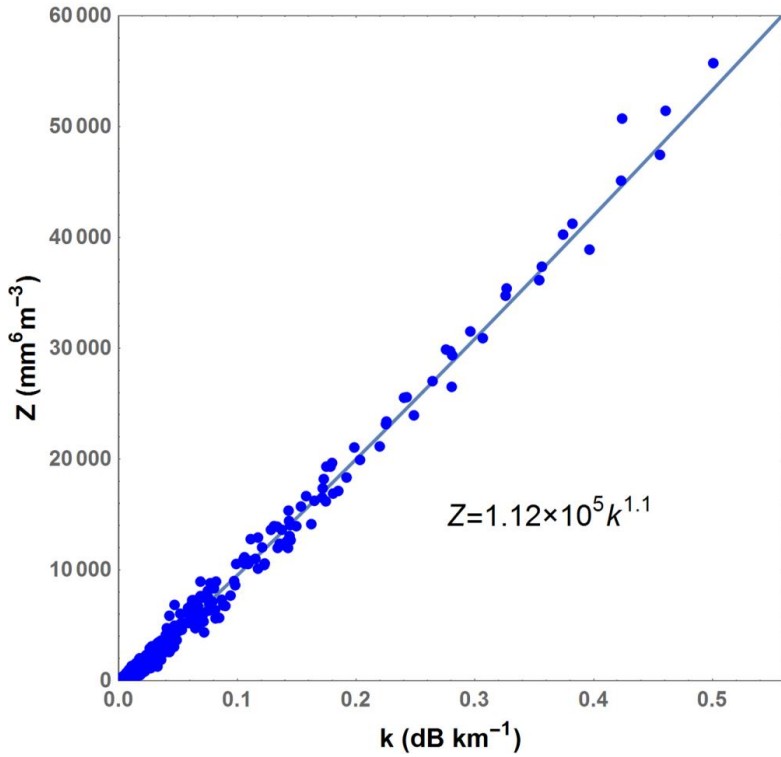


**Figure 4** *Z-k* relationship derived from DSD data using a non-linear power-law fit.





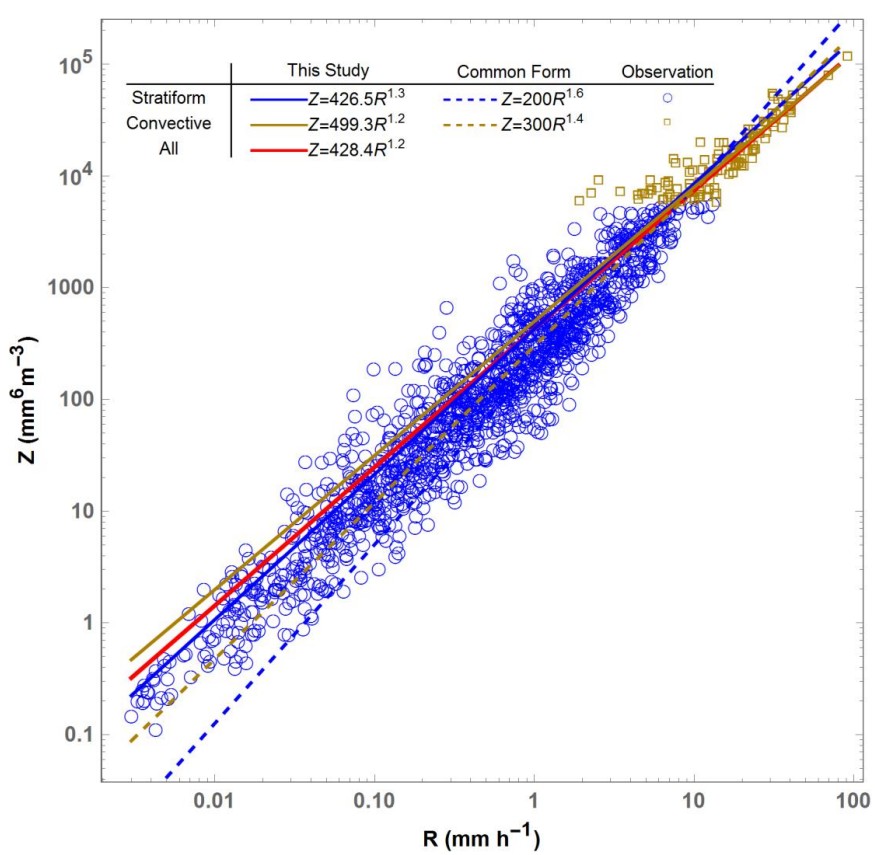


**Figure 5** *Z-R* relationships for different rainfall scenarios.









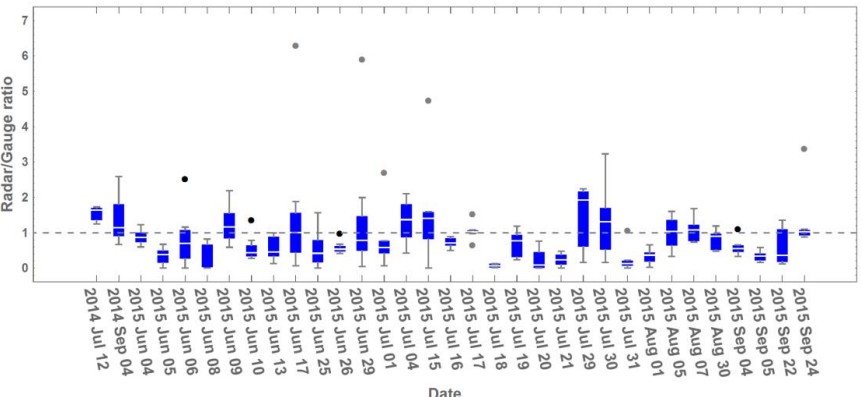

**Figure 6** Radar-gauge ratios of the daily accumulated rainfall for events covering at least 3 gauges. The dots denote the outlier values. Each box ranges from the 25th percentile to the 75th percentile with the middle line denoting the median value.

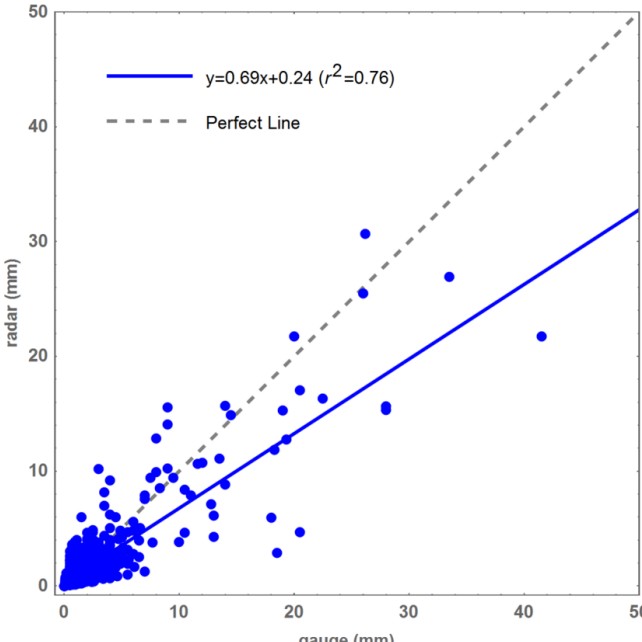

**Figure 7** The relationship of hourly rainfall accumulations from 8 rain gauges and the corresponding radar pixels for all the events.





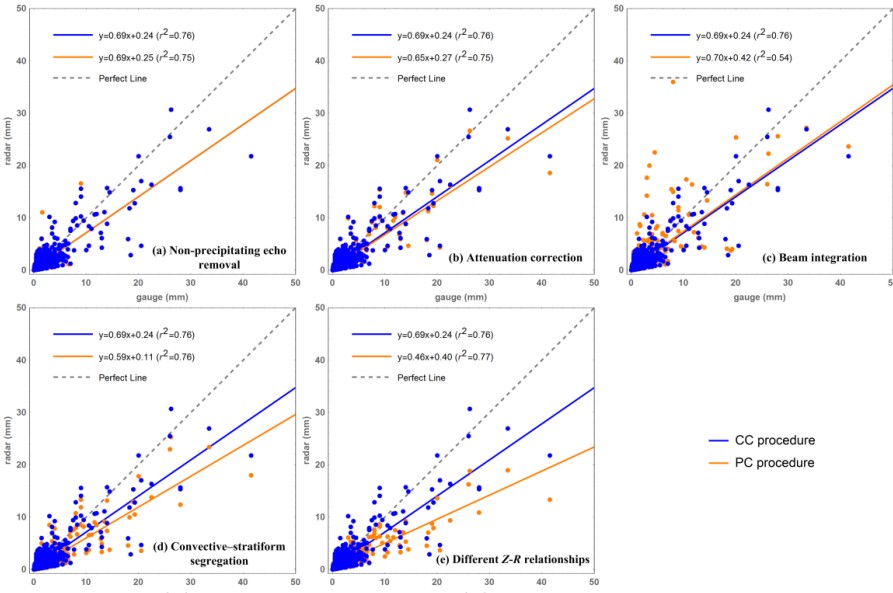

**Figure 8** Performance in radar-based rainfall estimation of different corrections: (a)

non-precipitation echo removal, (b) attenuation correction, (c) beam integration, (d)

convective–stratiform segregation, and (e) using different *Z-R* relationships for

converting the reflectivity to rainfall intensity. The blue and orange dots show the

results of complete-correction and partial-correction procedures, respectively.



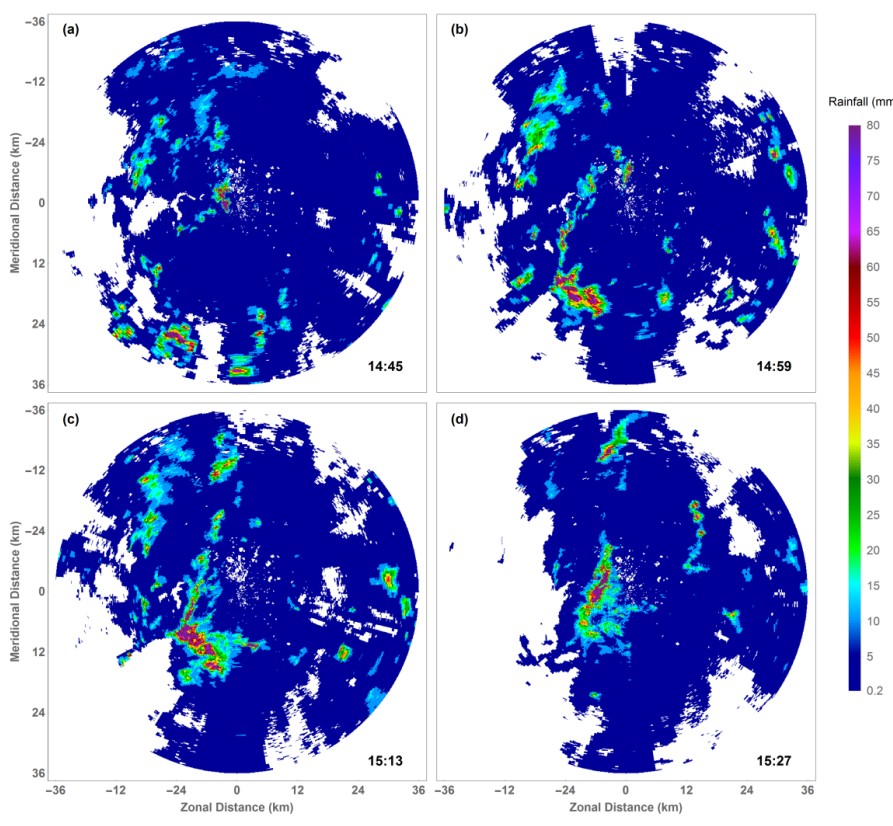


**Figure 9** Snapshots of the radar-based rainfall fields for the fast-moving rainfall event

of 4th September, 2015 at (a) 14:45, (b) 14:59, (c) 15:13 and (d) 15:27.






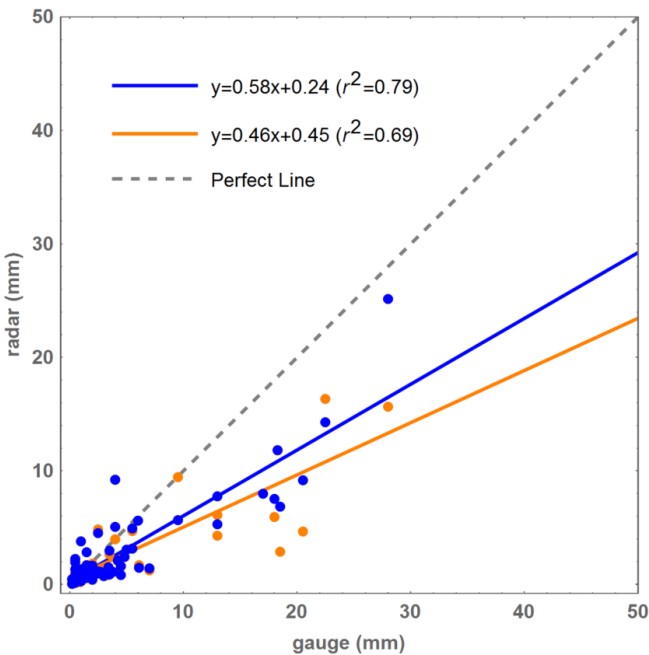


**Figure 10** Performance in radar-based rainfall estimation of the wind drift correction.

The blue and orange dots show the results of complete-correction and partial-correction

procedures, respectively.


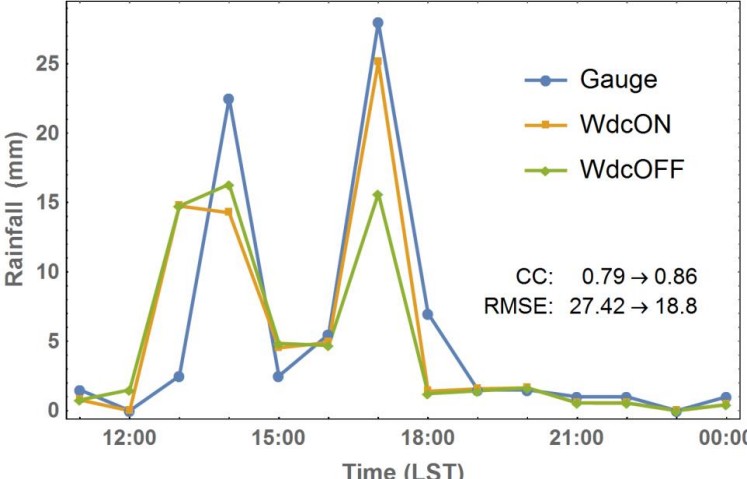


**Figure 11** Rain gauge, QPE with wind drifting correction and QPE without wind



drifting correction rainfall time series for fast-moving event, 4th-5th September 2015
at the position of closest gauge to radar.