# Peer review of "Exploring the potential of utilizing high resolution X-band radar for urban rainfall estimation"

_Atmospheric Measurement Techniques, 2016_

## Short Comment (SC1) · 15 Dec 2016

I have gone through this paper very quickly. I am far away from this topic but I really show much interesting it. It's a nice strategy to estimate rainfall. My question is what's the advangage of this technique compared with other much more cheap rainfall sensing technique. Anyway, I am far way from this topic, just absorbed by this technique.

---

## Referee Comment (RC1) · Anonymous Referee #1 · 19 Dec 2016

GENERAL COMMENT

This paper illustrates the processing of the observations collected by an X-band single-polarization radar in Beijing for hydrological purposes. Although the topic is of significant interest, the work is affected by a general lack of novelty (most of the employed procedures are well known and are here presented without a significant in-depth analysis) and serious theoretical flaws, in particular for the "wind drift" correction. In addition, the English language is not appropriate in many instances for a journal publication. The main issues are discussed more in detail below.

SPECIFIC COMMENTS

Radar calibration: calibration using a nearby disdrometer is actually a reasonable option, especially for longer wavelength radars (in the cited article, Lee and Zawadski

used S-band data). Indeed, at shorter wavelength such as X-band, in addition to path attenuation, the attenuation caused by the wet radome can induce serious underestimation of the reflectivity factor, up to several dB, e.g. Schneebeli and Berne (2012), Gorgucci et al. (2013), Frasier et al. (2013). Considering that the disdrometer in this study is very close to the radar, most of the measurements analyzed are likely coming from situations with rain over the radar also. This may explain the reported underestimation for higher reflectivity (>35 dBZ). Only qualitative results are reported in the manuscript, with figure 3 representing observations from a single event during a one-year period (by the way, I would exchange the x and y axes, since the disdrometer is the reference here). What about the other events and an overall quantitative evaluation of the calibration?

Beam integration: what is illustrated in this section appears to be a simple elevation selection, depending on the visibility. There is no mention of correction for partial beam blocking. If this is the case I think it may be simply called "beam selection", and should not be considered a correction procedure.

Local Z-R relations: the authors cite Steiner et al. (1995) work to differentiate rainfall type (convective/stratiform) based on a reflectivity threshold of 39 dBZ. However, the cited paper presents a more complex procedure based on the spatial structure of the reflectivity (intensity, peakedness,…). Steiner et al. report an overlap region between 20-35 dBZ, highlighting that "a simple reflectivity threshold method to separate convective from stratiform precipitation is insufficient". So, where does the 39 dBZ value comes from? Why do you need a different convective/stratiform partition method for the disdrometer data? Would it be possible to use the radar-based LWC method to select the corresponding disdrometer data for the separate Z-R retrievals? This may be more consistent, since in the end you need the Z-R relations for application to the radar observations.

Wind drift: the authors seem to confuse the motion vectors (advection of reflectivity patterns) and the wind vectors. At line 330 it is stated that "the advection velocity of

a rainy pixel (equal to the background wind velocity)". This is not true: the advection velocity is not the same as the wind velocity. Although a correlation may exist between storm advection and mid-tropospheric winds (e.g. Johns and Doswell, 1992; Kyznarova and Novak, 2005), the lower layers' winds (0-2 km) may actually dramatically differ from the advection motion. In addition, the low-level shear cannot be simply attributed to a velocity change (with constant direction), as reported in section 3.2. This is an over-simplification, not supported by neither theoretical arguments nor experimental evidence. It is also not clear why this "wind drift" correction is only shown for a single event, while the other corrections are applied to a bigger dataset. I'd rather suggest to carefully check the time synchronization between the radar and the gauge observations. In particular, which time was considered for the radar observations, since these are coming from different elevations (different scan time) depending on the azimuth sectors?

MINOR CORRECTIONS

- L. 18: "X-band-radar-based", too many hyphens. "X-band radar based" may read better.

- L. 23: "non-precipitation clutter" sounds tautological, it may be better to use something like "non-meteorological echoes (clutter)".

- L. 27: here and after: "distrometer", replace with "disdrometer".

- L. 56-58: it seems that a verb is missing (maybe replace "operating" with "operate").

- L. 57: replace "America" with "U.S.".

- L. 319 and 323: the reference to Caroline (2015) is missing.

- L.82-93: I'm not convinced that the wind drift effect should be considered an issue specific for X-band systems. While it is true that X-band have higher spatial resolution, due to the short range the height of the radar beam is in general lower, with a reduced impact of wind drift.

- L. 176: which kind of "prior knowledge" do you need for VPR? This is unclear.

- L. 279-280: "real-time atmospheric temperature profiles that is commonly used for convective-stratiform classification". Do you have a reference for this statement (convective-stratiform classification from temperature profiles)?

- L. 298: add references for the "standard" Z-R relations.

- L. 309: "distinguish" -> "differ"

- L. 318-327: the notation Delta_x may be confusing, since this usually indicates the zonal displacement.

- L. 622: "gian" -> "gain"

- Fig. 5: the mustard-colored and red lines have the same exponent (1.2) but different slopes in the plot. On the other hand, the blue and red lines show different exponents but seem to have the same slope. Looks like the coefficients are switched somewhere.

- Fig. 8: the result in panel (e) appears a bit counter-intuitive, since the "all" Z-R relation should over-estimate always respect the convective relation and also respect to the stratiform relation, for R higher than approx.. 1 mm/h. The scatterplot shows the opposite. This might be related with the Z-R coefficients issue (previous point).

- Fig. 9: if this is rainfall rate, the units should read "mm/h" instead of "mm".

REFERENCES

Schneebeli, M. and A. Berne, 2012: An Extended Kalman Filter Framework for Polarimetric X-Band Weather Radar Data Processing. J. Atmos. Oceanic Technol., 29, 711–730, doi: 10.1175/JTECH-D-10-05053.1.

Gorgucci, E., R. Bechini, L. Baldini, R. Cremonini, and V. Chandrasekar, 2013: The Influence of Antenna Radome on Weather Radar Calibration and Its Real-Time Assessment. J. Atmos. Oceanic Technol., 30, 676–689, doi: 10.1175/JTECH-D-12-00071.1.

Frasier, S., F. Kabeche, J. Figueras i Ventura, H. Al-Sakka, P. Tabary, J. Beck, and O. Bousquet, 2013: In-Place Estimation of Wet Radome Attenuation at X Band. J. Atmos. Oceanic Technol., 30, 917–928, doi: 10.1175/JTECH-D-12-00148.1.

Johns, R.; Doswell, C.A., III. Severe local storms forecasting. Weather Forecast. 1992, 7,588–612.

Kyznarova, H.; Novak, P. Development of Cell-Tracking Algorithm in the Czech Meteorological Institute. In Proceedings of WSN05—World Weather Research Programme —A Symposium on Nowcasting and Very Short Range Forecasting, Toulouse, France, 5-9 September 2005; p. 6.

---

## Referee Comment (RC2) · Anonymous Referee #2 · 12 Jan 2017

**Review of the paper**

**Exploring the potential of utilizing high resolution X-band radar for urban rainfall estimation**

By Wen-Yu Yang, Guang-Heng Ni, You-Cun Qi, Yang Hong, Ting Sun

No.: amt-2016-388

**General comments**

In this manuscpript, X-band radar observations are explored highlighting the strengths of high resolution rainfall estimations over urban area. A typical data processing for rain estimation that includes different corrections applied to X-band radar measurements are under analysis. The dataset used includes measurements from an X-band radar in single polarization installed in Beijing, a laser disdrometer (OTT Parsivel) and eight rain gauges installed in the area coverage by radar. The disdrometer measurements are used to obtain a relation for attenuation correction through reflectivity (calculated from DSDs) and two relations (R-Z) for rain estimation in convective and stratiform cases, while the rain gauges are used as reference for rain estimation. Several type of procedures for common error corrections are applied to one year of data, while one case study is selected to assess the wind drift correction.

This work is interesting, in fact, contributions to the potential of low-cost and small size radar, such as X-bands radar is an important task. However, this work lacks of novelty: most of the procedures used for common error corrections are derived from literature without improvements and validation. The main innovations of the work are related to the use of disdrometer measurements and the wind drift correction, although these procedures are not adequately described and the results are qualitative and not validated. Furthermore, possible interesting results, such as the calibration using the disdrometer and the wind drift correction are limited to one case study.

**Major comments**

- The use of disdrometer measurements for radar calibration in single polarization is an interesting approach since the DSDs measured are representative of the climatology of the region in which are collected. However, different points need to be clarified before applied this method: How the Mie calculation is performed? Which radar pixels are considered for the comparisons in Fig. 3? What is the error of the relationship shown in Fig.3? I suggest to investigate deeply the calibration results in particular, quantitative results, performances of the method and the extension at the entire dataset. These actions are indispensable before to decide if the calibration factor found is necessary or not to be applied.

- Certain issues (such as the instrumental error and the sampling error) have to be carefully considered when the disdrometer data have been used. Since in this work the disdrometer measurements are taken as reference, some considerations on instrumental limitations are needed. In relation to the attenuation correction: how is the performance of the relation between specific attenuation (k) and reflectivity shown in Fig.4? What indicate each point in the figure? Is the reflectivity at which time? How many radar volumes are plotted?

- The spectra of DSD collected by disdrometer have an error structure, being more or less sensitive to small drops or more precise for larger drops. Such errors impact applications, like the study of radar algorithms. Furthermore, some procedure of post processing for DSDs collected by disdrometer are necessary, for example to filter out spurious drops due to splashing or wind effect (Tokay et al, 2001). Furthermore, the R-Z relations obtained from the DSDs measurements need to be validate. In particular, the intrinsic validation (that can be obtained from the scatter plot between the Rain Rate (RR) derived from DSDs and the RR obtained from R-Z relation) and the comparison of rain with rain gauges.

- Besides the application of a fixed threshold (why 39 dBZ?) to divide stratiform/convective events a classification of rain regimes based on disdrometer measurements can be used (see Bringi et al 2003, Roberto et al 2016, Adirosi et al, 2015).

- In section *beam integration* the partial beam blocking is not correct, rather the elevation with optimal visibility is select without compensate the part of signal blocked. There are different approach to compensate the partial beam blocking effect for instance, that proposed by Bech et al 2003.

- The largest improvements found in the results shown in Tab. 3, are found for the beam integration procedure. This result appears obvious, if the radar beam is blocked the rain estimated by radar, compared to that measured by rain gauge will be underestimated. The correction that should be assessed is the rain estimated at the optimal visibility elevation before and after applied the partial beam blocking correction.

- In order to assess the improvements of Z-R relations from DSD measurements at least these validations are necessary: i) validate the performances of the Z-R relations in terms of intrinsic validation (as explained in previous comments) and ii) the performance of the Z-R relations applied to radar measurements comparing to the standard Z-R relations. In this

work the intrinsic validation is not implemented, while if the validation using radar measurements is applied or not is not clear (see lines 392-396 pag 20). I think that in this work is necessary a session dedicated to DSDs measured by Parsivel, that describes the Parsivel data processing and the validation of the relations derived (calibration, attenuation correction and R-Z relations).

**Minor Points**

Line 110 pag 7 Among the advanced methods that mitigate the error for rain estimation using X-band radar measurements, an approach that reduces the attenuation effect and calibration error is the combined algorithm between Kdp-R and Z-R (Vulpiani et Al. 2015).

Line 290 pag 15 What is "BW"?

Line 295 pag 15 Which is the origin of the threshold of 6.5 Kg $m^{-2}$ to categorize stratiform and convective pixels?

Lines 298 pag 16 Please insert the reference for the standard Z-R relationships.

Lines 326 pag 17 Since in this work the fall velocity is available from disdrometer measurements, did you check if the fixed falling speed of 5 m/s is representative of your case study? I suggest to use the fall velocity measured by disdrometer, at least for the pixels around the Parsivel in order to obtain the error on the hypothesis of fixed speed velocity.

Lines 237 pag 17. Do you mean Eq.(9) instead of Eq.(12)?

Lines 328-332 pag 17 Is not clear if the tracking algorithm is applied in this work? Please rewrite this part.

Lines 358-362 pag 19 Which are the radar pixels considered to calculate the *ra* and *ra*? Instead of select the event based on *ra* and *rd* ratio, you should associate a confidence level to each rain gauge based on the radar visibility and on the distance.

Line 366 pag 19 What about RMSE?

**References**

Tokay, A., A. Kruger, and W. Krajewski, 2001: Comparison of drop size distribution measurements by impact and optical disdrometers. J. Appl. Meteor., 40, 2083–2097.

Bringi, V. N., Chandrasekar, V., Hubbert, J., Gorgucci, E., Randeu, W. L., Schoenhuber M.: 9 Raindrop size distribution in different climatic regimes from disdrometer and dual-polarized 10 radar analysis, J. Atmos Sci. 60, 354-365, 2003.

Roberto, N., Adirosi, E., Baldini, L., Casella, D., Dietrich, S., Gatlin, P., Panegrossi, G., Petracca, M., Sanò, P., and Tokay, A.: Multi-sensor analysis of convective activity in central Italy during the HyMeX SOP 1.1, Atmos. Meas. Tech., 9, 535-552, doi:10.5194/amt-9-535-2016, 2016.

Adirosi, E., Baldini, L., Roberto, N., & Russo, F. (2016). C/S algorithm based on properties of dual-polarization radar measurements derived from disdrometer data. In *International Conference of Numerical Analysis and Applied Mathematics 2015, ICNAAM 2015.* (Vol. 1738). [430002] American Institute of Physics Inc.. DOI: 10.1063/1.4952215

Vulpiani, G., Baldini, L., and Roberto, N.: Characterization of Mediterranean hail-bearing storms using an operational polarimetric X-band radar, Atmos. Meas. Tech., 8, 4681-4698, doi:10.5194/amt-8-4681-2015, 2015.

Bech, J.; Codina, B.; Lorente, J. and Bebbington, D. The sensitivity of single polarization weather radar beam blockage correction to variability in the vertical refractivity gradient. *J. Atmos. Oceanic Technol.*, **2003**, 20, 845–855, doi: 10.1175/1520-0426(2003)020<0845:TSOSPW>2.0.CO;2

---

## Referee Comment (RC3) · Anonymous Referee #3 · 12 Jan 2017

The work explores common corrections that have to be applied to single polarization X-band Doppler weather radar measurements in order to get quantitative precipitation estimations (QPE). The test site is locate on Beijing (China) and one single polarization X-band radar, one distrometer (OTT Parsivel) and eight raingauges are available. Rainfall events from July 2014 to September 2015 (43) are considered, with focus on 4th September 2015.

Although the topic is currently very interesting, the work has a traditional approach, already deeply analyzed in several works (see Nielsen et al. 2013; Scheebeli et al., 2012; Pedersen et el., 2010 and their references). The first doubt is about the shortness of the observations: some uncertainties, like anomalous propagation, are directly linked to climatological conditions of the site that can not be evaluated with one year observations. According to the Reviewer #1, the wind drift correction contains severe

theoretical issues: it is arguable to apply advection-derived wind instead of wind profile below precipitation, obtained from observations or NWP. The corresponding strong assumption is that the convective system displacement corresponds at least to the wind in the atmosphere from the cloud base to the ground!

Moreover, the derivation of Z-R relationships discriminating stratiform and convective shows unclear key points: how is the 39 dBZ threshold chosen? How many data are respectively used for stratiform and convective non-linear fittings? Is the decimal precision of "a" coefficient in Z-R relationships (in this study 426.5 or 499.3) really meaningful? Which are data quality checks applied on distrometer? The two derived equations are quite similar (i.e. similar DSD for stratiform and convective rainfall), and it is quite surprising: how can the authors explain it?

To reduce the bias in radar-gauges comparison, the authors consider 33 of 43 events. This choice need to be clarified: how are they chosen? Which are their characteristics?

Finally, the authors assert that anomalous propagation (AP) contributes a minimal improvement. The effect of AP on weather radar measurements should be evaluated more rigorously: - the authors consider only rainfall events, while AP has impact also during dry weather, carrying to false rainfall; - the overall effect depends on AP climatology of the considered site, that must evaluated on longer period (see Bech et al 2012, ; Fornasiero et al, 2006a, 2006b).

The exposition of the study is unclear and very hard to evaluate. The language is often poor with several spelling mistakes, even in physical units ("Ghz" or "kw" in Table 1).

---

## Author Comment (AC1) · 7 Apr 2017

Response to Reviewer 1

We deeply appreciate the reviewer for his/her very insightful and constructive comments.

We would note that we decided to change our topic to "Utilizing X-band radar monitoring fast-moving rainfall events" considering the nature of the revision. Such change is motivated by following reasons: The urban hydrologic simulations are very sensitive to the spatiotemporal variability of rainfall (Schilling, 1991; Emmanuel,et al., 2012) and thus require rainfall inputs of high spatiotemporal resolution. Although X-band radars can provide rainfall products of high spatial resolution (Chen and Chandrasekar, 2015), they still lack the ability to provide products of high temporal resolution. The radarrainfall accumulations generated from periodic sampling often poorly represent the actual rain fields due to the coarse temporal resolution of the radar rainfall product. This error will be amplified for fast moving storms and fine spatial resolution data (Seo and Krajewski, 2015). In the revised manuscript, we monitor the fast-moving rainfall events with downscaled X-band radar product using the extrapolation technique. First, we quantitatively evaluate the "common error" correction approach to assess the quality of the coarse temporal resolution product. Then, we investigate the impacts of advection correction on the radar QPE. We also examine impacts of the physical factors on the correction accuracy.

The connection between the previous and revised manuscripts are: Same observations from the Beijing X-band radar system, including an X-band radar and a disdrometer; Same QPE algorithm to retrieve rainfall from radar measurement.

However, due to the unexpected amount of work in the revision, we are unable to finish the revision in time even though one extension had been kindly granted by the editor. As such, we first address the specific concerns of the reviewer as best as we can; meanwhile we are working on the revision with more thorough analysis.

Below we detail how we addressed the specific concerns of the reviewer: Major comments: Radar calibration: calibration using a nearby disdrometer is actually a reasonable option, especially for longer wavelength radars (in the cited article, Lee and Zawadski used S-band data). Indeed, at shorter wavelength such as X-band, in addition to path attenuation, the attenuation caused by the wet radome can induce serious underestimation of the reflectivity factor, up to several dB, e.g. Schneebeli and Berne (2012), Gorgucci et al. (2013), Frasier et al. (2013). Considering that the disdrometer in this study is very close to the radar, most of the measurements analyzed are likely coming from situations with rain over the radar also. This may explain the reported underestimation for higher reflectivity (>35 dBZ). Only qualitative results are reported in the manuscript, with figure 3 representing observations from a single event during a one year period (by the way, I would exchange the x and y axes, since the disdrometer is the reference here). What about the other events and an overall quantitative evaluation of the calibration? Response: We thank the reviewer for the suggestion. The reviewer's analysis of underestimation is very insightful and we will add a detailed analysis in the revised manuscript. Also, we agree with reviewer that a single event comparison is not comprehensive. In the revised manuscript, four fast-moving events are selected to evaluate the performance of X-band radar QPE. Therefore, the evaluation will be done for the four selected fast-moving events to make the calibration more convincing.

Beam integration: what is illustrated in this section appears to be a simple elevation selection, depending on the visibility. There is no mention of correction for partial beam blocking. If this is the case I think it may be simply called "beam selection", and should not be considered a correction procedure. Response: Corrected as suggested.

Local Z-R relations: the authors cite Steiner et al. (1995) work to differentiate rainfall type (convective/stratiform) based on a reflectivity threshold of 39 dBZ. However, the cited paper presents a more complex procedure based on the spatial structure of the reflectivity (intensity, peakedness). Steiner et al. report an overlap region between 20-35 dBZ, highlighting that "a simple reflectivity threshold method to separate convective from stratiform precipitation is insufficient". So, where does the 39 dBZ value comes from? Why do you need a different convective/stratiform partition method for the disdrometer data? Would it be possible to use the radar-based LWC method to select the corresponding disdrometer data for the separate Z-R retrievals? This may be more consistent, since in the end you need the Z-R relations for application to the radar observations.. Response: We thank the reviewer for pointing out our incorrect citation of Steiner et al., (1995). The 39 dBZ threshold was used just as a first order estimation. In the previous manuscript, we used this simple threshold method due to its computational efficiency compared with the radar-based LWC method. In the revised manuscript, as we now focus on only four rainfall events, it is feasible to use the radar-based LWC method.

Wind drift: the authors seem to confuse the motion vectors (advection of reflectivity patterns) and the wind vectors. At line 330 it is stated that "the advection velocity of a rainy pixel (equal to the background wind velocity)". This is not true: the advection velocity is not the same as the wind velocity. Although a correlation may exist between storm advection and mid-tropospheric winds (e.g. Johns and Doswell, 1992; Kyznarova and Novak, 2005), the lower layers' winds (0-2 km) may actually dramatically differ from the advection motion. In addition, the low-level shear cannot be simply attributed to a velocity change (with constant direction), as reported in section 3.2. This is an over-simplification, not supported by neither theoretical arguments nor experimental evidence. It is also not clear why this "wind drift" correction is only shown for a single event, while the other corrections are applied to a bigger dataset. I'd rather suggest to carefully check the time synchronization between the radar and the gauge observations. In particular, which time was considered for the radar observations, since these are coming from different elevations (different scan time) depending on the azimuth sectors? Response: We thank the reviewer for pointing out our confusion between the motion vectors and the wind vectors. In fact we intended to investigate the temporal sampling bias caused by advection rather than the wind drift effects (Thorndahl et al., 2017). It has been acknowledged that radar-rainfall accumulations generated by weather radars from periodic sampling often incorrectly represent actual rain fields. Coarse temporal resolution radar product suffers spatially discontinuous patterns that were caused by the intermittent radar scanning frequency. This error will be amplified for fast moving storms and fine spatial resolution data (Seo and Krajewski, 2015). Therefore, in the revised manuscript, we use an extrapolation techniques to downscale the radar product to very fine temporal resolution (1 min). The effect of temporal sampling error on the results will be further discussed for four fast moving events.

Minor comments: L. 319 and 323:: the reference to Caroline (2015) is missing. Response: We apologize that the Caroline (2015) should be Sandford (2015). The reference is in line 577

L.82-93: I'm not convinced that the wind drift effect should be considered an issue specific for X-band systems. While it is true that X-band have higher spatial resolution, due to the short range the height of the radar beam is in general lower, with a reduced impact of wind drift. Response: We confused the wind velocity and advection velocity in the previous manuscript. Now in the revised manuscript, we have removed the part on wind drift effect and instead investigate the temporal sampling bias caused by advection.

L. 176: which kind of "prior knowledge" do you need for VPR? This is unclear Response: The prior knowledge is the variability of vertical structure of precipitation over Beijing. Based on the vertical structure, we can convert the reflectivity at a high elevation to relatively lower altitude. The vertical structure of precipitating system can be well resolved by the remote sensing instruments. Compared to ground-based radar, space-borne radar has great advantages in measuring the vertical structure of storm thanks to the less interference from the earth curvature, mountain blockage, and beam broadening.

"real-time atmospheric temperature profiles that is commonly used for convective-stratiform classification". Do you have a reference for this statement (convective-stratiform classification from temperature profiles)? Response: This statement can be found in Qi et al., (2013). They use the reflectivity at the altitude of -10° for the convective-stratiform classification.

L. 318-327: the notation Delta_x may be confusing, since this usually indicates the zonal displacement. Response: The notation $\Delta x$ has been modifed as $\Delta s$.

Fig. 5: the mustard-colored and red lines have the same exponent (1.2) but different slopes in the plot. On the other hand, the blue and red lines show different exponents but seem to have the same slope. Looks like the coefficients are switched somewhere Response: We thank the reviewer for pointing out our mistake in the plot. We confuse the coefficients of "all" Z-R relation and Stratiform Z-R relation in this figure.

The revised figure is shown as fig 1. Despite the above analysis, since our convective/stratiform partition method is insufficient, the Z-R relation for both stratiform and convective rainfall is inappropriate, that means we still need to replot the figure in our revised manuscript.

Fig. 8: the result in panel (e) appears a bit counter-intuitive, since the "all" Z-R relation should over-estimate always respect the convective relation and also respect to the stratiform relation, for R higher than approx.. 1 mm/h. The scatterplot shows the opposite. This might be related with the Z-R coefficients issue (previous point). Response: We have double-checked that the plot is correct based on the Z-R relationships used in the previous manuscript. The reviewer suggested that "all" Z-R relation should over-estimate both respect the convective relation and also respect to the stratiform relation. However, from the fig 2, we can clearly see that "all" Z-R underestimate the respect the convective relation. Therefore, using the "all" relation will underestimate the convective events like what is shown in our fig 8. Despite the above reason, since our convective/stratiform partition method is insufficient, the Z-R relation for both stratiform and convective rainfall is inappropriate, that means we still need to replot the figure in our revised manuscript.

All the other minor comments have been corrected as suggested

References: Chen, H., Chandrasekar, V.: The quantitative precipitation estimation system for Dallas–Fort Worth (DFW) urban remote sensing network, J. Hydrol., 531, 259-271, 2015. Emmanuel, I., Andrieu, H., Leblois, E., Flahaut, B.: Temporal and spatial variability of rainfall at the urban hydrological scale, J. Hydrol., 430, 162–172, 2012. Fabry, F., Bellon, A., Duncan, M.R., Austin, G.L: High resolution rainfall measurements by radar for very small basins: the sampling problem reexamined, J. Hydrol., 161, 415–428, 1994. Qi, Y., Zhang, J., Zhang, P.: A real-time automated convective and stratiform precipitation segregation algorithm in native radar coordinates, Q. J. R. Meteorolog. Soc., 139, 2233-2240, 2013. Schilling, W.: Rainfall data for urban hydrology: what do we need, Atmos. Res., 27, 5–21, 1991. Seo, B., Krajewski, W.F.:

Correcting temporal sampling error in radar-rainfall: Effect of advection parameters and rain storm characteristics on the correction accuracy. J. Hydrol., 531, 272–283, 2015. Thorndahl, S., Einfalt, T., Willems, P. et al.: Weather radar rainfall data in urban hydrology, Hydrol. Earth Syst. Sci., 21, 1359–1380, 2017

Please also note the supplement to this comment:
http://www.atmos-meas-tech-discuss.net/amt-2016-388/amt-2016-388-AC1-supplement.pdf

---

## Author Comment (AC2) · 7 Apr 2017

Response to Reviewer 2

We deeply appreciate the reviewer for his/her very insightful and constructive comments.

We would note that we decided to change our topic to "Utilizing X-band radar monitoring fast-moving rainfall events" considering the nature of the revision. Such change is motivated by following reasons: 1) The urban hydrologic simulations are very sensitive to the spatiotemporal variability of rainfall (Schilling, 1991, Emmanuel,et al 2012) and thus require rainfall inputs of high spatiotemporal resolution. Although X-band radars can provide rainfall products of high spatial resolution (Chen and Chandrasekar, 2015), they still lack the ability to provide products of high temporal resolution. 2) The radarrainfall accumulations generated from periodic sampling often poorly represent the actual rain fields due to the coarse temporal resolution of the radar rainfall product. This error will be amplified for fast moving storms and fine spatial resolution data (Seo and Krajewski, 2015). In the revised manuscript, we monitor the fast-moving rainfall events with downscaled X-band radar product using the extrapolation technique. First, we quantitatively evaluate the "common error" correction approach to assess the quality of the coarse temporal resolution product. Then, we investigate the impacts of advection correction on the radar QPE. We also examine impacts of the physical factors on the correction accuracy.

The connection between the previous and revised manuscripts are: 1. Same observations from the Beijing X-band radar system, including an X-band radar and a disdrometer; 2. Same QPE algorithm to retrieve rainfall from radar measurement.

However, due to the unexpected amount of work in the revision, we are unable to finish the revision in time even though one extension had been kindly granted by the editor. As such, we first address the specific concerns of the reviewer as best as we can; meanwhile we are working on the revision with more thorough analysis.

Below we detail how we addressed the specific concerns of the reviewer: Major comments: 1. The use of disdrometer measurements for radar calibration in single polarization is an interesting approach since the DSDs measured are representative of the climatology of the region in which are collected. However, different points need to be clarified before applied this method: How the Mie calculation is performed? Which radar pixels are considered for the comparisons in Fig. 3? What is the error of the relationship shown in Fig.3? I suggest to investigate deeply the calibration results in particular, quantitative results, performances of the method and the extension at the entire dataset. These actions are indispensable before to decide if the calibration factor found is necessary or not to be applied. Response: We thank the reviewer for the suggestions. The details of Mie calculation can be found in www.ou.edu/radar/module01radarApps.pdf. The error may come from several sources:

1) The attenuation caused by the wet radome: since the disdrometer in our study is very close to the radar, most of the measurements analyzed are likely coming from situations with rain over the radar also; 2) Different sampling methods: radar performs the volumetric measurement while disdrometer conducts measurement at the point scale; 3) The vertical variability of reflectivity. A deeper analysis of the fast-moving events is conducted in the revised manuscript.

2. Certain issues (such as the instrumental error and the sampling error) have to be carefully considered when the disdrometer data have been used. Since in this work the disdrometer measurements are taken as reference, some considerations on instrumental limitations are needed. In relation to the attenuation correction: how is the performance of the relation between specific attenuation (k) and reflectivity shown in Fig.4? What indicate each point in the figure? Is the reflectivity at which time? How many radar volumes are plotted? Response: In the revise manuscript, a comparison between disdrometer and gauges will be added. And as state above, there will be a deep analysis for the fast-moving events in the revised manuscript. In the previous manuscript, each point in the figure indicates 5 min averaged Z and k calculated based on disdrometer data. The correlation coefficient of the regression is 0.99. Actually, this figure is based on one-year measurements of disdrometer from July 2014 to September 2015. There is no radar data used here.

3. The spectra of DSD collected by disdrometer have an error structure, being more or less sensitive to small drops or more precise for larger drops. Such errors impact applications, like the study of radar algorithms. Furthermore, some procedure of post processing for DSDs collected by disdrometer are necessary, for example to filter out spurious drops due to splashing or wind effect (Tokay et al, 2001). Furthermore, the R-Z relations obtained from the DSDs measurements need to be validate. In particular, the intrinsic validation (that can be obtained from the scatter plot between the Rain Rate (RR) derived from DSDs and the RR obtained from R-Z relation) and the comparison of rain with rain gauges. Response: Such comparison will be conducted in the revised

manuscript.

4. Besides the application of a fixed threshold (why 39 dBZ?) to divide stratiform/convective events a classification of rain regimes based on disdrometer measurements can be used (see Bringi et al 2003, Roberto et al 2016, Adirosi et al, 2015).

Response: We thank the reviewer for pointing out our incorrect citation of Steiner et al. (1995). In the previous manuscript, we used this simple threshold method due to its computational efficiency compared with the radar-based LWC method. In the revised manuscript, as we now focus on only four rainfall events, it is feasible to use the radar-based LWC method.

5. In section beam integration the partial beam blocking is not correct, rather the elevation with optimal visibility is select without compensate the part of signal blocked. There are different approach to compensate the partial beam blocking effect for instance, that proposed by Bech et al 2003.

Response: We thank the reviewer for the suggestion. As Reviewer 1 pointed out, what we did was a simple elevation selection depending on the visibility rather than beam integration. Therefore, partial beam blocking correction is not considered in this manuscript,

6. The largest improvements found in the results shown in Tab. 3, are found for the beam integration procedure. This result appears obvious, if the radar beam is blocked the rain estimated by radar, compared to that measured by rain gauge will be underestimated. The correction that should be assessed is the rain estimated at the optimal visibility elevation before and after applied the partial beam blocking correction..

Response: We thank the reviewer for the suggestion. As stated in Response 5, we have realized what we did was a simple elevation selection. However, what we compared is not rainfall estimated based on the blocked radar beam. Instead, what we compared is rainfall estimated based on the single lowest elevation without beam blockage

(i.e., 4.0 ° in this study). Both the rainfall estimated by beam selection and the single lowest elevation are not impacted by beam blockage. Also, as the theme of the revised manuscript is changed, this part will be removed.

7. In order to assess the improvements of Z-R relations from DSD measurements at least these validations are necessary: i) validate the performances of the Z-R relations in terms of intrinsic validation (as explained in previous comments) and ii) the performance of the Z-R relations applied to radar measurements comparing to the standard Z-R relations. In this work the intrinsic validation is not implemented, while if the validation using radar measurements is applied or not is not clear (see lines 392-396 pag 20). I think that in this work is necessary a session dedicated to DSDs measured by Parsivel, that describes the Parsivel data processing and the validation of the relations derived (calibration, attenuation correction and R-Z relations)..

Response: We thank the reviewer for the suggestion. We fully agree that an intrinsic validation is necessary and will be added in the revised manuscript.

Minor comments: 1. Line 110 pag 7 Among the advanced methods that mitigate the error for rain estimation using X-band radar measurements, an approach that reduces the attenuation effect and calibration error is the combined algorithm between Kdp-R and Z-R (Vulpiani et Al. 2015). Response: Corrected as suggested.

2. Line 290 pag 15 What is "BW"? Response: BW is the angular width of the radar beam between the half-power points (for Beijing radar the value is 0.9°). This information has been added in the revised manuscript in line 294.

3. Line 295 pag 15 Which is the origin of the threshold of 6.5 Kg m-2 to categorize stratiform and convective pixels? Response: The threshold 6.5 kg m-2 comes from Qi et al. (2013).

4. Lines 298 pag 16 Please insert the reference for the standard Z-R relationships Response: The references (Marshall and Palmer 1948, Fulton et al., 1998) have been

added in the revised manuscript.

5. Lines 326 pag 17 Since in this work the fall velocity is available from disdrometer measurements, did you check if the fixed falling speed of 5 m/s is representative of your case study? I suggest to use the fall velocity measured by disdrometer, at least for the pixels around the Parsivel in order to obtain the error on the hypothesis of fixed speed velocity. Response: In fact we intended to investigate the temporal sampling bias caused by advection rather than the wind drift effects (Thorndahl et al. 2017). For the advection correction, the assumption of fall velocity is not needed in the revised manuscript.

6. Lines 237 pag 17. Do you mean Eq.(9) instead of Eq.(12)? Response: Yes, we meant eq. (9) rather than eq. (12). And we thank the reviewer for pointing out our incorrect reference.

7. Lines 328-332 pag 17 Is not clear if the tracking algorithm is applied in this work? Please rewrite this part. Response: In the previous manuscript, the tracking algorithm is used to calculate the advection velocity. In the revised manuscript, this part is modified as follows: "The nowcasting algorithm in this study will be used to compute advection velocity vector and the rainfall trend; then, based on the linear interpolation of the computed velocity vector and the rainfall trend, rain rate maps with 1 min resolution will be generated; finally, rainfall accumulation using an increased number of rain rate maps will be computed and compared against the gauge data.

8. Lines 358-362 pag 19 Which are the radar pixels considered to calculate the rd and ra? Instead of select the event based on ra and rd ratio, you should associate a confidence level to each rain gauge based on the radar visibility and on the distance. Response: We thank the reviewer for the suggestion and will conduct such analysis in the revised manuscript. However, we did NOT use the ra and rd ratios to select the rainfall events for analysis; instead, we use these ratios to assess the performance of the X-band radar QPE system.

9. Line 366 pag 19 What about RMSE? Response: The RMSE is 2.1 mm h-1. It has been added in line 369 of the revised manuscript.

References: References: Chen, H., Chandrasekar, V.: The quantitative precipitation estimation system for Dallas–Fort Worth (DFW) urban remote sensing network, J. Hydrol., 531, 259-271, 2015. Emmanuel, I., Andrieu, H., Leblois, E., Flahaut, B.: Temporal and spatial variability of rainfall at the urban hydrological scale, J. Hydrol., 430, 162– 172, 2012. Fabry, F., Bellon, A., Duncan, M.R., Austin, G.L: High resolution rainfall measurements by radar for very small basins: the sampling problem reexamined, J. Hydrol., 161, 415–428, 1994. Qi, Y., Zhang, J., Zhang, P.: A real-time automated convective and stratiform precipitation segregation algorithm in native radar coordinates, Q. J. R. Meteorolog. Soc., 139, 2233-2240, 2013. Schilling, W.: Rainfall data for urban hydrology: what do we need, Atmos. Res., 27, 5–21, 1991. Seo, B., Krajewski, W.F.: Correcting temporal sampling error in radar-rainfall: Effect of advection parameters and rain storm characteristics on the correction accuracy. J. Hydrol., 531, 272–283, 2015. Thorndahl, S., Einfalt, T., Willems, P. et al.: Weather radar rainfall data in urban hydrology, Hydrol. Earth Syst. Sci., 21, 1359–1380, 2017

Please also note the supplement to this comment:
http://www.atmos-meas-tech-discuss.net/amt-2016-388/amt-2016-388-AC2-supplement.pdf

---

## Author Comment (AC3) · 7 Apr 2017

Response to Reviewer 3

We deeply appreciate the reviewer for his/her very insightful and constructive comments.

We would note that we decided to change our topic to "Utilizing X-band radar monitoring fast-moving rainfall events" considering the nature of the revision. Such change is motivated by following reasons: 1) The urban hydrologic simulations are very sensitive to the spatiotemporal variability of rainfall (Schilling, 1991, Emmanuel,et al, 2012) and thus require rainfall inputs of high spatiotemporal resolution. Although X-band radars can provide rainfall products of high spatial resolution (Chen and Chandrasekar, 2015), they still lack the ability to provide products of high temporal resolution. 2) The radar-

rainfall accumulations generated from periodic sampling often poorly represent the actual rain fields due to the coarse temporal resolution of the radar rainfall product. This error will be amplified for fast moving storms and fine spatial resolution data (Seo and Krajewski, 2015). In the revised manuscript, we monitor the fast-moving rainfall events with downscaled X-band radar product using the extrapolation technique. First, we quantitatively evaluate the "common error" correction approach to assess the quality of the coarse temporal resolution product. Then, we investigate the impacts of advection correction on the radar QPE. We also examine impacts of the physical factors on the correction accuracy.

The connection between the previous and revised manuscripts are: 1. Same observations from the Beijing X-band radar system, including an X-band radar and a disdrometer; 2. Same QPE algorithm to retrieve rainfall from radar measurement.

However, due to the unexpected amount of work in the revision, we are unable to finish the revision in time even though one extension had been kindly granted by the editor. As such, we first address the specific concerns of the reviewer as best as we can; meanwhile we are working on the revision with more thorough analysis.

Below we detail how we addressed the specific concerns of the reviewer: Major comments: 1. The first doubt is about the shortness of the observations: some uncertainties, like anomalous propagation, are directly linked to climatological conditions of the site that can not be evaluated with one year observations. Response: We thank the reviewer for the suggestion. And we agree with the reviewer one year's observations cannot be used to evaluate the effects of anomalous propagation. As the theme of the revised manuscript is changed, this part will be removed.

2. The wind drift correction contains severe theoretical issues: it is arguable to apply advection-derived wind instead of wind profile below precipitation, obtained from observations or NWP. The corresponding strong assumption is that the convective system displacement corresponds at least to the wind in the atmosphere from the cloud base to

the ground! Response: We did make oversimplified assumption in the wind drift correction. In fact we intended to investigate the temporal sampling bias caused by advection rather than the wind drift effects (Thorndahl et al. 2017). It has been acknowledged that radar-rainfall accumulations generated by weather radars from periodic sampling often incorrectly represent actual rain fields. Coarse temporal resolution radar product suffers spatially discontinuous patterns that were caused by the intermittent radar scanning frequency. This error will be amplified for fast moving storms and fine spatial resolution data (Seo and Krajewski, 2015). Therefore, in the revised manuscript, we use an extrapolation techniques to downscale the radar product to very fine temporal resolution (1 min). The effect of temporal sampling error on the results will be further discussed for four fast moving events.

3. Moreover, the derivation of Z-R relationships discriminating stratiform and convective shows unclear key points: how is the 39 dBZ threshold chosen? How many data are respectively used for stratiform and convective non-linear fittings? Is the decimal precision of "a" coefficient in Z-R relationships (in this study 426.5 or 499.3) really meaningful? Which are data quality checks applied on distrometer? The two derived equations are quite similar (i.e. similar DSD for stratiform and convective rainfall), and it is quite surprising: how can the authors explain it? Response: We thank the reviewer for pointing out our mistake. Here, we apologize that we make a serious mistake which is citing wrong paper. In the previous manuscript, we used this simple threshold method due to its computational efficiency compared with the radar-based LWC method. In the revised manuscript, as we now focus on only four rainfall events, it is feasible to use the radar-based LWC method.

4. To reduce the bias in radar-gauges comparison, the authors consider 33 of 43 events. This choice need to be clarified: how are they chosen? Which are their characteristics? Response: In this work, 8 gauges are used to validate the radar QPE. Among the 43 events, there are 33 events during which at least three gauges have valid measurements. Therefore these 33 events are chosen to investigate the radar-gauge ratios

of the daily accumulated rainfall (Fig. 6). When comparing hourly accumulated rainfall, all the events in the study period are utilized.

5. Finally, the authors assert that anomalous propagation (AP) contributes a minimal improvement. The effect of AP on weather radar measurements should be evaluated more rigorously: - the authors consider only rainfall events, while AP has impact also during dry weather, carrying to false rainfall; - the overall effect depends on AP climatology of the considered site, that must evaluated on longer period (see Bech et al 2012, ; Fornasiero et al, 2006a, 2006b). Response: We thank the reviewer for the suggestion. As stated in Response 1, we have realized one year's observation cannot be used to evaluate the effects of anomalous propagation. Also, as the theme of the revised manuscript is changed, this part will be removed.

6. The exposition of the study is unclear and very hard to evaluate. The language is often poor with several spelling mistakes, even in physical units ("Ghz" or "kw" in Table 1). Response: We will thoroughly polish the language and correct all the technical issues.

References: Chen, H., Chandrasekar, V.: The quantitative precipitation estimation system for Dallas–Fort Worth (DFW) urban remote sensing network, J. Hydrol., 531, 259-271, 2015. Emmanuel, I., Andrieu, H., Leblois, E., Flahaut, B.: Temporal and spatial variability of rainfall at the urban hydrological scale, J. Hydrol., 430, 162–172, 2012. Fabry, F., Bellon, A., Duncan, M.R., Austin, G.L: High resolution rainfall measurements by radar for very small basins: the sampling problem reexamined, J. Hydrol., 161, 415–428, 1994. Qi, Y., Zhang, J., Zhang, P.: A real-time automated convective and stratiform precipitation segregation algorithm in native radar coordinates, Q. J. R. Meteorolog. Soc., 139, 2233-2240, 2013. Schilling, W.: Rainfall data for urban hydrology: what do we need, Atmos. Res., 27, 5–21, 1991. Seo, B., Krajewski, W.F.: Correcting temporal sampling error in radar-rainfall: Effect of advection parameters and rain storm characteristics on the correction accuracy. J. Hydrol., 531, 272–283, 2015. Thorndahl, S., Einfalt, T., Willems, P. et al.: Weather radar rainfall data in urban

hydrology, Hydrol. Earth Syst. Sci., 21, 1359–1380, 2017

Please also note the supplement to this comment:
http://www.atmos-meas-tech-discuss.net/amt-2016-388/amt-2016-388-AC3-supplement.pdf

---

## Author Comment (AC4) · 7 Apr 2017

In general, there are several ways that can measure rainfall such as rain gauge, radar, satellite, and so on. The standard way of measuring rainfall is rain gauge which measure rainfall at or near the ground. It can accurately measure the total rainfall in a certain interval of time. However, rain gauge is point scale measurement which means it cannot effectively describe the spatial variability of rainfall.

Weather radar measures the energy returned from a precipitation target and then transform it into rain rate. Compared with rain gauges, weather radars can do continuous measurements and provide high spatiotemporal resolution (e.g. 1 km/5–10 min) rainfall data that allow hydrological simulation to be conducted at very fine scales. Moreover, the spatial resolution of X-band radar can be as fine as $\sim$100 m, which makes X-band

radar more suitable to monitor the highly heterogeneous rainfall in urban area.